# Dysregulation of the mRNA Expression of Human Renal Drug Transporters by Proinflammatory Cytokines in Primary Human Proximal Tubular Epithelial Cells

**DOI:** 10.3390/pharmaceutics16020285

**Published:** 2024-02-16

**Authors:** Yik Pui Tsang, Tianran Hao, Qingcheng Mao, Edward J. Kelly, Jashvant D. Unadkat

**Affiliations:** Department of Pharmaceutics, University of Washington, Seattle, WA 98195, USA; a6245912@uw.edu (Y.P.T.); tihao@uw.edu (T.H.); edkelly@uw.edu (E.J.K.)

**Keywords:** proinflammatory cytokines, renal drug transporters, renal clearance, disease-drug interaction, proximal tubular epithelial cells

## Abstract

Proinflammatory cytokines, which are elevated during inflammation or infections, can affect drug pharmacokinetics (PK) due to the altered expression or activity of drug transporters and/or metabolizing enzymes. To date, such studies have focused on the effect of cytokines on the activity and/or mRNA expression of hepatic transporters and drug-metabolizing enzymes. However, many antibiotics and antivirals used to treat infections are cleared by renal transporters, including the basal organic cation transporter 2 (OCT2), organic anion transporters 1 and 3 (OAT1 and 3), the apical multidrug and toxin extrusion proteins 1 and 2-K (MATE1/2-K), and multidrug resistance-associated protein 2 and 4 (MRP2/4). Here, we determined the concentration-dependent effect of interleukin-6 (IL-6), IL-1β, tumor necrosis factor (TNF)-α, and interferon-γ (IFN-γ) on the mRNA expression of human renal transporters in freshly isolated primary human renal proximal tubular epithelial cells (PTECs, *n* = 3–5). PTECs were exposed to either a cocktail of cytokines, each at 0.01, 0.1, 1, or 10 ng/mL or individually at the same concentrations. Exposure to the cytokine cocktail for 48 h was found to significantly downregulate the mRNA expression, in a concentration-dependent manner, of OCT2, the organic anion transporting polypeptides 4C1 (OATP4C1), OAT4, MATE2-K, P-glycoprotein (P-gp), and MRP2 and upregulate the mRNA expression of the organic cation/carnitine transporter 1 (OCTN1) and MRP3. OAT1 and OAT3 also appeared to be significantly downregulated but only at 0.1 and 10 ng/mL, respectively, without a clear concentration-dependent trend. Among the cytokines, IL-1β appeared to be the most potent at down- and upregulating the mRNA expression of the transporters. Taken together, our results demonstrate for the first time that proinflammatory cytokines transcriptionally dysregulate renal drug transporters in PTECs. Such dysregulation could potentially translate into changes in transporter protein abundance or activity and alter renal transporter-mediated drug PK during inflammation or infections.

## 1. Introduction

The effects of inflammation caused by acute infection on the human pharmacokinetics (PK) of drugs are well documented [1,2]. These changes in drug PK are mainly attributed to pathophysiological increases in plasma cytokine concentrations, resulting in changes in the functional activity of hepatic drug-metabolizing enzymes (e.g., cytochrome P450, CYPs) or transporters. For instance, influenza B infection resulted in significant elevation of theophylline plasma concentrations in children, resulting in severe toxicity [3], likely due to the downregulation of hepatic CYP1A2 (which metabolizes theophylline) by cytokines [4]. Drug PK can also be altered by inflammation chronic infections and autoimmune diseases [1]. These chronic inflammatory states are maladaptive and are characterized by the elevation of cytokines in the plasma for prolonged periods, albeit often to a lesser extent than those seen in acute scenarios [4]. For example, plasma concentrations of S-verapamil in patients with active Crohn’s disease were significantly higher than in healthy individuals. Since S-verapamil is metabolized by multiple CYP enzymes, prolonged exposure to cytokines in these patients likely decreased the expression of these enzymes, resulting in an eight-fold increase in the S-verapamil area under the plasma concentration–time curve (AUC) [5].

Inflammation can also influence the PK of drugs that are substrates of drug transporters. Drug transporters are integral membrane proteins that facilitate the influx and/or efflux of molecules (drugs or endogenous compounds) across cell membranes. Those located in the membranes of hepatocytes, enterocytes, and renal proximal tubular epithelial cells (PTECs) are the main determinants of absorption or disposition. In critically ill patients with sepsis, atorvastatin’s plasma AUC and maximal plasma concentration (C_max_) were 18- and 15-fold increased relative to healthy volunteers [6]. The hepatic clearance of atorvastatin is driven by its uptake into hepatocytes via organic anion-transporting polypeptides (OATP1B1/3), where it is completely metabolized by CYP3A4/5 [7]. However, in this study, since atorvastatin’s plasma concentrations were not altered in the presence of co-administered CYP3A4 inhibitors, the drastic increase in atorvastatin exposure due to sepsis was likely due to the downregulation of OATPs. Le Vée et al. investigated the effects of proinflammatory cytokines interleukin (IL)-1β, IL-6, and tumor necrosis factor (TNF)-α on the expression of hepatic drug transporters in human hepatocytes. They reported significant downregulation of the mRNA expression of a wide variety of efflux and uptake transporters, including P-glycoprotein (P-gp), multidrug resistance-associated protein 2 (MRP2), breast cancer resistance protein (BCRP), organic cation transporter 1 (OCT1), sodium taurocholate co-transporting polypeptide (NTCP), as well as OATP1B1, 1B3, and 2B1 [8,9].

The major proinflammatory cytokines with elevated circulating concentrations during inflammatory conditions are IL-1β, IL-6, TNF-α, and interferon (IFN)-γ (Table 1, [10,11,12,13,14]). In human kidneys, the basolateral uptake of drugs is primarily facilitated by organic anion transporter 1–3 (OAT1-3), OCT2, and OATP4C1, while the apical efflux of drugs is mainly driven by multidrug and toxin extrusion protein 1 and 2-K (MATE1/2-K) and MRP2/4. Although many antibiotics and antivirals used to treat infections (e.g., amoxicillin and tenofovir) are cleared by the kidneys through active secretion [15,16], only one study has determined the in vivo effect of inflammation on renal drug transporter function [17]. Moreover, there are no in vitro studies on the effect of cytokines on renal transporter expression or function. Here, we used freshly isolated PTECs from human kidneys to systematically determine the impact of the above four major proinflammatory cytokines at their pathophysiological plasma concentrations, either individually or in combination (henceforth referred to as ‘cocktail’), on the mRNA expression of major human renal transport proteins.

## 2. Materials and Methods

### 2.1. Chemicals and Reagents

Dulbecco’s Modified Eagle’s Medium/Nutrient Mixture F-12 powder without glucose was purchased from United States Biological (Salem, MA, USA). D-(+)-Glucose, HEPES (4-(2-hydroxyethyl)-1-piperazineethanesulfonic acid), and sodium bicarbonate were purchased from MilliporeSigma (Bellevue, WA, USA). Roswell Park Memorial Institute 1640 media (RPMI 1640), Dulbecco’s Phosphate-Buffered Saline (DPBS), Insulin-Transferrin-Selenium (100X) (100X ITS-G), Antibiotic–Antimycotic (100X) (100X AA), collagenase type IV and Dispase^®^ powder was purchased from Gibco (Billings, MT, USA). For this study, 24-well collagen I-coated plates were purchased from Corning (Corning, NY, USA). Percoll^®^ density gradient media was purchased from Cytiva (Marlborough, MT, USA). Recombinant human cytokines (IL-1β, IL-6, TNF-α, and IFN-γ) were purchased from R&D Systems (Minneapolis, MN, USA).

### 2.2. Cell Isolation and Culture

Human PTECs were isolated from normal human kidneys (GFR > 60 mL/min) sourced from the US Organ Procurement Organizations (OPOs) via Novabiosis, Inc. (Durham, NC, USA) and were identified as non-transplantable or surgical discards from healthy adult donors (see Table 2 for demographics). The kidneys were preserved in sterile UW solution at 4 °C with a clamp time to isolation of less than 36 h. Under sterile conditions, the kidneys were decapsulated, and the cortex slices were dissected and minced into approximately 1 mm^3^ pieces. Every 5 g of minced tissue was subjected to enzymatic digestion in 50 mL of sterile-filtered solution containing 0.75 mg/mL collagenase type IV and 0.75 mg/mL dispase in DPBS. The suspension was gently agitated for 2 h at 37 °C on an orbital shaker. Then, the cell suspension was passed through a 100 μm cell strainer (to remove the undigested tissue), and the filtrate was centrifuged at 200× *g* for 7 min. This and all subsequent centrifugation and handling steps were performed at 4 °C. The resulting cell pellets were combined and twice washed with cold RPMI 1640 media. After washing, the cell pellet was resuspended in cold RPMI 1640 media, and the suspension was loaded on a discontinuous Percoll gradient with a density of 1.03 g/mL and 1.07 g/mL in 12–16 15 mL Falcon tubes and centrifuged at 1600× *g* for 25 min to isolate the PTECs. After centrifugation, the PTEC layer at the intersection of the two layers of different densities was extracted, pooled, and washed twice with cold RPMI 1640 media. The resulting cell pellet was resuspended in warm PTEC media (DMEM/F12 media with low glucose (1 g/L), supplemented with 10 mM HEPES, 14 mM sodium bicarbonate, 1X ITS-G and 1X AA). The cells were then diluted to a density of 600,000 cells/mL and plated (0.5 mL/well) in 24-well collagen I-coated plates.

### 2.3. Proinflammatory Cytokine Treatments

Recombinant human cytokines (IL-1β, IL-6, TNF-α, and IFN-γ) were reconstituted as per the manufacturer’s instructions. Briefly, the lyophilized powder for each cytokine was first reconstituted individually at 100 μg/mL in DPBS and subsequently aliquoted in small volumes to minimize freeze–thaw cycles. These 100 μg/mL stock solutions of individual cytokines were further serially diluted in DPBS to make four separate working stock concentrations: 10 μg/mL, 1000 ng/mL, 100 ng/mL, and 10 ng/mL. For the cytokine cocktail, the working stock solutions of each cytokine were diluted to 1:1000 in the PTEC media (0.4% DPBS *v*/*v*) to achieve the final combined concentration ranges of each cytokine (10 ng/mL, 1 ng/mL, 0.1 ng/mL, or 0.01 ng/mL). For the individual cytokines, the working stock solutions of each concentration were similarly diluted to 1:1000 in PTEC media, with additional DPBS added to ensure that its final concentration matched that in the cytokine cocktail incubation media. Once the PTECs were plated and allowed to adhere for 24 h in culture, the media was replaced with either PTEC media with 0.4% DPBS (vehicle control) or PTEC media containing the cytokines (individual or cocktail). PTECs from each donor (*n* = 3–5) were incubated in triplicate for 48 h with the cytokines, and the respective media were replaced every 24 h. Cytokine depletion in the media at 24 h was monitored using ELISA kits from R&D Systems (Minneapolis, MN, USA).

### 2.4. RNA Isolation, cDNA Synthesis and Quantification via Real-Time Quantitative PCR (qPCR)

At the end of the cytokine treatments, the total RNA was isolated from the PTECs using the RNeasy^®^ Mini Kit from Qiagen (Valencia, CA, USA). The concentration of the isolated RNA was determined using a NanoDrop^TM^ fluroimeter (ThermoFisher Scientific, Waltham, MA, USA) and normalized to the lowest concentration in each batch in order to normalize the cDNA amount produced and check for the stability of the housekeeping gene.

The extracted RNA was converted into complementary DNA (cDNA) using the High-Capacity cDNA Reverse Transcription^®^ Kit (Applied Biosystems, Foster City, CA, USA). Every 20 μL of reverse transcription reaction contained 2 μL of 10X RT Buffer, 0.8 μL of 25× dNTP Mix (100 nM), 2 μL of 10× Random Primer mix, 1 μL of MultiScribe^TM^ Reverse Transcriptase, 3.2 μL of RNase-free water, and 10 μL of the isolated RNA. The cDNA synthesis reaction was performed with CFX96 real-time system-C1000 Thermal Cycler (Bio-Rad Laboratories, Hercules, CA, USA) with the following conditions: 25 °C for 10 min, 37 °C for 120 min, and 85 °C for 5 min.

The relative quantification of gene expression consisted of simultaneous amplification of the target gene product and the selected endogenous housekeeping gene product glyceraldehyde-3-phosphate dehydrogenase (GAPDH), using TaqMan^TM^ assays, TaqMan^TM^ Fast Advanced Master Mix for qPCR (Applied Biosystems, Foster City, CA, USA), and the CFX96 real-time system-C1000 Thermal Cycler (Bio-Rad Laboratories, Hercules, CA, USA). Each treatment condition was analyzed in TempPlate^®^ Semi-Skirted 96-well PCR plates (USA Scientific, Inc., Ocala, FL, USA) in triplicate. Each reaction was performed in singleplex following the protocol provided by the manufacturer. The TaqMan^TM^ Gene Expression Assays used were as follows: Hs01010726_m1 for OCT2 (SLC22A2), Hs00537914_m1 for OAT1 (SLC22A6), Hs00198527_m1 for OAT2 (SLC22A7), Hs01056646_m1 for OAT3 (SLC22A8), Hs00698884_m1 for OATP4C1 (SLCO4C1), Hs00945829_m1 for OAT4 (SLC22A11), Hs00268200_m1 for organic cation/carnitine transporter 1 (OCTN1) (SLC22A4), Hs00929869_m1 for OCTN2 (SLC22A5), Hs00217320_m1 for MATE1 (SLC47A1), Hs00945652_m1 for MATE2-K (SLC47A2), Hs00184500_m1 for P-gp (ABCB1), Hs01053790_m1 for BCRP (ABCG2), Hs00960489_m1 for MRP2 (ABCC2), Hs00978452_m1 for MRP3 (ABCC3), Hs00988721_m1 for MRP4 (ABCC4), and Hs02786624_g1 for GAPDH (GAPDH). Each 20 μL reaction contained 10 μL of the 2× TaqMan^TM^ Fast Advanced Master Mix, 1 μL of the 20× TaqMan^TM^ Gene Expression Assay, and 9 μL of cDNA (approximately 30 ng of cDNA) diluted in RNase-free water. The thermocycling conditions were 2 min at 50 °C, 20 s at 95 °C, followed by 40 cycles for 3 s at 95 °C and 30 s at 60 °C.

### 2.5. Data and Statistical Analysis

The relative quantification of the mRNA expression in PTECs from each donor was determined using the comparative cycle threshold (C_t_) method, i.e., 2^−∆∆Ct^, where ∆∆C_t_ = ∆C_t_ of each treated sample—mean ∆C_t_ of the vehicle-treated controls, and ∆C_t_ = C_t_ of the gene of interest—C_t_ of GAPDH, the housekeeping gene [18]. Gene expressions with C_t_ values above 36 were excluded from analyses as they were considered to be low and, therefore, unreliable. The mean expression, relative to vehicle control, for each treatment was averaged across the donors (n = 3–5) and analyzed using one-way analysis of variance (ANOVA) followed by post hoc Fisher’s LSD test to correct for multiple comparisons. A p-value less than 0.05 was considered statistically significant. All data analyses were performed using GraphPad Prism 10 (GraphPad Software, La Jolla, CA, USA).

## 3. Results

### 3.1. Cytokine Cocktail Significantly and Differentially Dysregulated the mRNA Expression of Renal Drug Transporters

Following exposure to the cytokine cocktail, the mRNA expression of OATP4C1, OCT2, MATE2-K, OAT4, and P-gp was significantly downregulated in a concentration-dependent manner by up to 67%, 68%, 77%, 66%, and 50%, respectively (Figure 1A,C). MRP2 was also downregulated in a concentration-dependent manner but only up to 1 ng/mL (Figure 1C). In contrast, the mRNA expression of MRP3 and OCTN1 was induced in a concentration-dependent manner by up to ~2.5- and ~4-fold, respectively (Figure 1B,D). OAT1 and OAT3 appeared to be significantly downregulated by the cytokine cocktail by 50% at 0.1 ng/mL and 10 ng/mL, respectively. However, there was no clear concentration dependency, and there was considerable variability among the donors. The mRNA expression of the remaining transporters was not significantly affected by the cytokine cocktail at any concentration. In addition, after 24 h, the concentrations of IL-1β, TNF-α, and IFN-γ in the media were depleted by approximately 40%. In contrast, IL-6 was secreted by the PTECs, resulting in a seven-fold higher concentration at 24 h compared to the initial nominal treatment concentration.

### 3.2. Individual Cytokines Significantly and Differentially Dysregulated the mRNA Expression of Renal Drug Transporters

To determine which cytokine(s) are the major perpetrators of the effect observed with the cocktail, the effect of the individual cytokines (i.e., IL-1β, IL-6, TNF-α, and IFN-γ) on the mRNA expression of the renal drug transporter was investigated. In general, amongst the four cytokines, IL-1β appeared to be the most potent at downregulating the mRNA expression of OCT2, OATP4C1, OAT4, MATE2-K, and MRP2 and upregulating the mRNA expression of MRP3 and OCTN1 (Figure 2A–D). IFN-γ also appeared to downregulate OCT2 and P-gp, but to a much lesser extent (Figure 2A,D). For these transporters, the effects of cytokines seem to be concentration-dependent. IL-6, though not significant, overall appears to be the cytokine responsible for the slight downregulation of the mRNA expression of OAT1, OAT3, and BCRP by the cytokine cocktail (Figure 2A,B). Table 3 provides a summary of cytokine-induced changes in transporter mRNA expression at 10 ng/mL of individual cytokine or cocktail treatment (unless otherwise indicated). 

## 4. Discussion

To the best of our knowledge, this is the first study that has used freshly isolated PTECs from human kidneys to investigate transcriptional dysregulation of renal drug transporters caused by four major proinflammatory cytokines (i.e., IL-1β, IL-6, TNF-α, and IFN-γ) individually or in combination as a cocktail. These cytokines were selected due to their pivotal role in mediating innate immune response and because they are significantly and simultaneously elevated in a myriad of infectious and autoimmune diseases [10,11,12,13,14,19,20,21,22]. The cytokine concentrations were chosen to encompass those observed in plasma during inflammation (Table 1). In addition, we also studied the effect of the individual cytokines at the concentrations used in the cocktail to identify the perpetrator within the cocktail.

Of the basal renal transporters studied, only the mRNA expression of OCT2 and OATP4C1 was downregulated by the cytokine cocktail in a concentration-dependent manner. For these two transporters, variability in gene regulation between lots was considerably lower than other transporters. In contrast, the mRNA expression of OAT1 and OAT3 was significantly downregulated (by ~50%) by the 0.1 ng/mL and 10 ng/mL cocktail, respectively. However, the data were much more variable between donors, with some showing potent downregulation of mRNA expression while others showing little to no changes in expression at the same cytokine concentrations (Appendix A). This difference between the results for mRNA expression of OAT1/3 versus OCT2/OATP4C1 suggests differences in their regulatory machinery. In addition, there could also be genetic polymorphisms of these transporters among the donors that resulted in different basal mRNA expression and possibly different extent of regulation by the cytokines [23,24]. Nonetheless, these findings are in line with the in vivo data in rats published by Pour et al., where polyinosinic:polycytidilic acid (poly I:C) intraperitoneal injection (causing inflammation) downregulated the mRNA expression of Oct2, Oatp4c1, Oat1 and 3 [25]. Additionally, the only in vivo human study that investigated the impact of inflammation on the activity of renal transporters showed that the clearance of furosemide, an OAT1/3 in vivo probe drug, was significantly reduced by ~40% in pregnant women during acute pyelonephritis vs. after resolution of pyelonephritis (i.e., paired study), indicating an inflammation-induced downregulation of OAT1/3 activity [17].

Among the apical renal transporters, the mRNA expression of OAT4, MATE2-K, P-gp, and MRP2 was significantly downregulated by the cytokine cocktail (only up to 1 ng/mL for MRP2) in a concentration-dependent manner. In line with our results, the mRNA expression of rat P-gp and Mrp2 is decreased during inflammation [25,26]. However, no such data are available for MATE2-K and OAT4. Interestingly, both these studies report that the greater the downregulation of the expression of Mate1, the more abundant MATE isoform in the kidney. This was not observed in our study and was potentially due to interspecies differences in how MATE1 is regulated during inflammation. In contrast, the expression of OCTN1 in PTECs was significantly induced by the cytokine cocktail exposure in a concentration-dependent manner. Maeda et al. also showed such induction in a human fibroblast-like synoviocyte cell line (MH7A) exposed to IL-1β and TNF-α at 1 and 10 ng/mL, respectively [27]. In contrast, in the above study by Pour et al., they reported a repression of rat Octn1 mRNA by the cytokines [25].

To identify the perpetrator of the effect observed with the cytokine cocktail, we determined the effect produced by the individual cytokines. Among the four cytokines tested, IL-1β appeared to be the most potent in regulating the mRNA expression of OCT2, OATP4C1, OAT4, OCTN1, MATE2-K, MRP2, and MRP3. For the remaining transporters, no significant effect was elicited by the individual cytokines. There was no apparent synergism between the tested cytokines, and, similar to the cocktail data, considerable variability was also observed between donors in response to the individual cytokines. The concentrations of cytokines used in this study were not sufficient to capture the half-maximal effect concentrations (ED_50_) for each transporter, and further research is needed to evaluate if the cytokines act synergistically or additively to regulate renal drug transporters.

The above findings on the regulation of both basal and apical renal transporters during inflammation have significant clinical implications. Both OCT2 and MATE2-K transport and eliminate organic cationic drugs from the blood into the urine [28], including metformin, an oral hypoglycemic drug primarily used in the treatment of type-2 diabetes mellitus. Since metformin exhibits in vitro and in vivo anti-inflammatory effects in both animals and humans [29,30], it could potentially be incorporated into therapies for treating several inflammatory diseases [30]. In the presence of inflammation, the clearance of metformin may be reduced due to downregulation of OCT2. Additionally, downregulation of MATE2-K on the apical side of PTECs could further lead to intracellular accumulation of metformin, as well as other drugs transported by the renal organic cation transport system. Similarly, the downregulation of MRP2 on the apical side could potentially lead to the accumulation of methotrexate in the PTECs, which could result in renal toxicity [31]. For OATP4C1 and P-gp, since cardiac glycosides are narrow therapeutic window drugs [32], the downregulation of these two transporters during inflammation could lead to reduced renal clearance of these drugs, resulting in elevated plasma concentrations and potentially off-target toxicity of these drugs. For the apical transporter OAT4, due to its bidirectionality, its downregulation could decrease or increase systemic drug exposure depending on the drug substrate. Interestingly, while increases in the plasma concentrations of some drugs during inflammation may be undesirable, the downregulation of basal transporters or the upregulation of apical transporters during inflammation could serve as a protective mechanism against drug-induced nephrotoxicity [33]. 

Information on the mechanism(s) by which transporters are regulated by cytokines is sparse. At the transcriptional level, many endo- and xenobiotic transporters are regulated by transcriptional factors [34]. Many putative binding sites for stress-related transcription factors are found on the promoter sequences of BCRP, P-gp, and MRP genes [35]. Indeed, the Janus kinase/signal transducer and activator of transcription (JAK/STAT), mitogen-activated protein kinase/extracellular regulated kinase (MAPK/ERK), and phosphoinositide 3 kinase (PI3K)/protein kinase B (AKT) pathways have all been shown to be involved in the acute regulation of hepatic drug transporters by proinflammatory cytokines in mice and human hepatocytes. These signals work by altering the expression of nuclear hormone receptors, including the pregnane X receptor (PXR), the constitutive androstane receptor (CAR), and the retinoid X receptor (RXR) [36,37,38,39,40,41]. Interestingly, IL-1β and TNF-α have been shown to transcriptionally regulate the expression of OCTN1 in MH7A cells via the nuclear factor kappa B (NF-κB), a transcriptional factor downstream of the PI3K/AKT pathway [27]. Thus, it is possible that IL-1β regulates the mRNA expression of transporters in PTECs via the same mechanism. In sum, our data provide important information to design future studies aimed at elucidating the underlying mechanisms through which cytokines regulate renal drug transporters.

There are a few limitations to our study. First, although the PTECs were freshly isolated from human kidneys, the mRNA expression of some transporters, particularly the basolateral uptake transporters, decayed rapidly in culture (Appendix A). As a result, the expression of these transporters relative to each other was significantly different compared to those measured in human kidney tissue [42,43]. Most notably, the mRNA expression of OAT2 and OAT1/3 dropped to unquantifiable levels after 2 and 7 days in culture, respectively. For this reason, mRNA expression data of OAT2 are not reported. Where mRNA expression was quantifiable, the degradation of mRNA over time was accounted for by expressing all the cytokine treatment data relative to the control treatment. Therefore, we were able to interpret any changes in mRNA expression caused by the cytokines. Second, our PTEC model needs to be optimized so that the mRNA expression can be quantified over a longer period of time and to determine if changes in transporter mRNA expression translate into changes in activity. This is important as changes in mRNA expression do not always result in changes in the transporter’s protein expression and activity [44]. For example, OATs and OCT2 are known to undergo post-translational modifications that result in changes in their activities [45]. Although we attempted to measure the effect of cytokine exposure on OAT1/3 and OCT2 transporter activity, the basal activities of these transporters could not be detected. Currently, aside from the standard culture format, there are multiple other kidney models of PTECs that have demonstrated functional OAT activity [46]. However, they are lower in throughput and more costly. Notably, an in vitro model with PTECs cultured on Transwell© inserts has demonstrated inhibitable OAT and OCT activity [47]. Unfortunately, since proinflammatory cytokines disrupt the formation of tight junctions [48,49,50], this model cannot be used in this study. Third, for efficiency purposes, we need to optimize methods to cryopreserve the PTECs. We did test if they could be cryopreserved, but upon thawing and plating, the mRNA expression of transporters was found to be lower than that in fresh cells for the same duration in culture, and there was no measurable OAT1/3 and OCT2 activity. Lastly, we observed changes in cytokine concentrations in culture over 24 h, which included a reduction in IL-1β, TNF-α, and IFN-γ, along with the secretion of IL-6 by the PTECs. While this suggests that the cytokine exposure over the 24 h period might differ from the initial concentrations, it does not affect the interpretation of our data as the cytokine concentrations are still within the pathophysiological ranges. Overall, despite these limitations, our data provide a basic understanding of how the mRNA expression of drug transporters is altered by proinflammatory cytokines.

In summary, we demonstrated for the first time that proinflammatory cytokines (at pathophysiological concentrations) significantly dysregulated the mRNA expression of several renal drug transporters in PTECs. The cytokine cocktail significantly downregulated the mRNA expression of OCT2, OATP4C1, OAT4, MATE2-K, P-gp, and MRP2 and upregulated the mRNA expression of OCTN1 and MRP3 in a concentration-dependent manner. We note here that this is the first time that cytokines have been reported to upregulate the mRNA expression of human renal transporters. Among the cytokines included in the cocktail, IL-1β was found to be the major perpetrator of this regulation. This transcriptional dysregulation should translate into changes in transporter protein abundance and activity, resulting in altered renal transporter-mediated drug PK. This work provides foundational data for future work on the underlying mechanisms and magnitude of this dysregulation. 

## Figures and Tables

**Figure 1 pharmaceutics-16-00285-f001:**
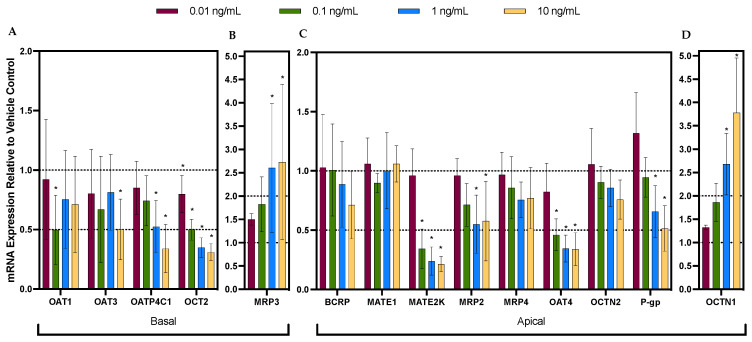
Concentration-dependent effect of cytokine cocktail (IL-6, IL-1β, TNF-α, and IFN-γ) on the mRNA expression of major basal (**A**,**B**) and apical (**C**,**D**) renal drug transporters in proximal tubular epithelial cells (PTECs). Each cocktail contained all four individual cytokines at 0.01 (purple bar), 0.1 (green bar), 1 (blue bar), or 10 ng/mL (yellow bar). Data are mean ± S.D. of 3–5 donors, each conducted in triplicate. Data are expressed relative to the mRNA expression in the PTECs treated with the vehicle control. Statistical significance (* *p* ≤ 0.05) was determined using a one-way analysis of variance (ANOVA) followed by post hoc Fisher’s least significant difference (LSD) test for pairwise comparisons between the treatment groups and the vehicle control. The quantification of the expression of the gene of interest was calculated using the comparative cycle threshold (C_t_) method using the formula 2^−∆∆Ct^ [18]. Glyceraldehyde-3-phosphate dehydrogenase (GAPDH) was used as the housekeeping gene. Data for organic anion transporter 2 (OAT2) are not included since its mRNA expression was not quantifiable (C_t_ values > 36).

**Figure 2 pharmaceutics-16-00285-f002:**
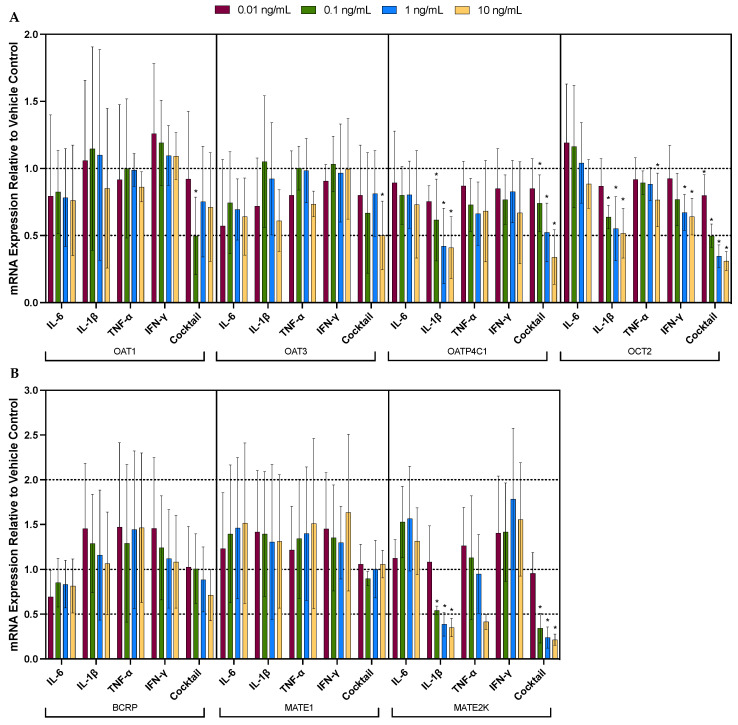
The effect of individual cytokines (i.e., IL-1β, IL-6, TNF-α, and IFN-γ) on the mRNA expression of renal drug transporters (**A**) OAT1/3, OCT2, OATP4C1; (**B**) BCRP, MATE1, MATE2K; (**C**) MRP2/3/4; (**D**) OAT4, OCTN1/2, P-gp. Each individual cytokine was tested at 0.01 (purple bar), 0.1 (green bar), 1 (blue bar), and 10 ng/mL (yellow bar). For ease of comparison, the corresponding concentration cocktail data (from Figure 1) are included. Data are mean ± S.D. of 3–5 donors, each conducted in triplicate. Data are expressed relative to the mRNA expression in the vehicle control. Statistical significance (* *p* ≤ 0.05) was determined using ANOVA followed by post hoc Fisher’s LSD test to make pairwise comparisons between the treatment groups and vehicle control. The expression of the gene of interest for each donor was calculated using the comparative C_t_ method with the formula 2^−∆∆Ct^ [18]. GAPDH was used as the housekeeping gene. Data for OAT2 are not included since its mRNA expression was not quantifiable (C_t_ values > 36).

**Table 1 pharmaceutics-16-00285-t001:** Range of plasma concentration (in pg/mL) of proinflammatory cytokines (interleukin-6 (IL-6), IL-1β, tumor necrosis factor (TNF)-α, and interferon-γ (IFN-γ)) in patients with inflammation and in healthy individuals.

Condition	IL-6	IL-1β	TNF-α	IFN-γ	Group Size (N)	References
COVID-19	1.6–4823.0 [10]	1.6–8.3 [10]	0.8–112.4 [10]	2.7–434.7 [11]	1959 [10], 63 [11]	[10,11]
HIV-1 infection	1.5–933.2	1.5–2751.2	1.5–389.1	1.5–2511.9	120	[12]
Malaria	1.2–812.8	10.2–812.83	1.1–44.7	1.5–309.0	113	[13]
Healthy adults under the age of 45	0.16–37.7	0.17–24.0	0.93–26.8	0.14–126.8	55	[14]

**Table 2 pharmaceutics-16-00285-t002:** Demographic information of the kidney donors.

Donor ID	Age	Body Mass Index (BMI)	Gender	Race	Cause of Death	Alcohol Use	Tobacco Use	Substance Use
AJJM288 (D3)	47	26.45	Female	African American	Stroke	No	No	No
AJLA067 (D4)	34	34.9	Male	African American	Anoxia	No	No	No
AJLL353 (D6)	48	28.36	Female	Caucasian	Stroke	No	Yes	No
AKAT206 (D7)	31	24.56	Female	Caucasian	Anoxia	No	No	No
AKAG138 (D9)	22	22.5	Female	Caucasian	Anoxia	No	No	No

**Table 3 pharmaceutics-16-00285-t003:** Summary of the effect on mRNA expression of transporters after 48 h of treatment of PTECs with individual cytokines or the cytokine cocktail (10 ng/mL, unless stated otherwise). Values shown are percent downregulated (↓), percent upregulated (↑), or no significant change (↔) relative to the control treatment.

Basolateral Transporters	IL-6	IL-1β	TNF-α	IFN-γ	Cocktail
OAT1	↔				↓ 50% ^a^
OAT3	↔				↓ 50%
OATP4C1	↔	↓ 59%	↔	↔	↓ 66%
OCT2	↔	↓ 48%	↔	↔	↓ 69%
Apical Transporters					
BCRP	↔				
MATE1	↔				
MATE2-K	↔	↓ 65%	↓ 59%	↔	↓ 78%
MRP2	↔	↓ 50%	↔	↔	↓ 45% ^b^
MRP3	↔	↑ 152% ^b^	↔	↔	↑ 173%
MRP4	↔				
OAT4	↔	↓ 64%	↔	↔	↓ 66%
OCTN1	↔	↑ 235%	↔	↔	↑ 278%
OCTN2	↔				
P-gp	↔	↔	↔	↓ 32%	↓ 49%

Maximum change was observed at ^a^ 0.1 ng/mL or ^b^ at 1 ng/mL.

## Data Availability

Data are contained within the article.

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
