# Peer review of "Dysregulation of the mRNA Expression of Human Renal Drug Transporters by Proinflammatory Cytokines in Primary Human Proximal Tubular Epithelial Cells"

_pharmaceutics, 2024, doi:10.3390/pharmaceutics16020285_

Round 1

Reviewer 1 Report

Comments and Suggestions for Authors

The authors study the dysregulation of the mRNA expression of human renal drug transporters by proinflammatory cytokines in primary human proximal tubular epithelial cells. Although the fact that this paper first describe the effect of cytokines individuals or in cocktail on the expression of drug transporters, it may be many limits.

Major comments :

- the authors present only results on the mRNA expression of drug transporters but it is very important to see if this regulation are sufficient to modulate the activity of these transporters. So, it lacks of activity measurements or protein level detection.

- The authors must present the basal mRNA level of expression of these transporters in cell controls when they begin the treatement and when they finish the treatment. In fact, the basal expression of transporters can decrease because there is a dedifferenciation of the cells along the culture. It presents many limits to have only data on mRNA expression.

Minor comments

In the introduction, the authors can add other references about cytokine effects on hepatic drug transporters, notably the work by Le Vee et al team.

Reviewer 2 Report

Comments and Suggestions for Authors

This manuscript describes the impact of proinflammatory cytokines on transcription of drug transport proteins in cultured renal tubular epithelial cells. The manuscript is well-written and interesting, and the work is well done. My criticisms are minor and are mostly related to grammar; there is excessive use of commas throughout the manuscript and occasional other grammatical errors. The manuscript should have a grammar edit before publication. Overall, I believe this work is a valuable addition to the literature on transport proteins.

Specific comments are below:

Introduction

Overall, the introduction is clear and indicates the relevance of the study topic.

Line 40: The paper cited describes changes in theophylline clearance in children infected with influenza B (not A as stated), please correct this. It also might be more clear to write this sentence as: “For instance, influenza B infection resulted in significant elevation of theophylline….” .

Line 54: “…18- and 15-fold of that in healthy volunteers” would be better written: “…18- and 15-fold increased relative to in healthy volunteers.”

Line 64: “…cleared by renal excretion including secretion…” is confusing as worded. Please reword.

Line 71: This would be more clear if you change “…major human renal transporters.” to “major human renal transport proteins.”

Table 1: Can you add the group sizes (N) for each condition to the table?

Materials and Methods:

The methods are clear and well-written.

Line 116: I believe “cytokine6” is a typo.

Results:

The results are well-displayed in the figures. Please see individual figure comments.

Figure 1: The figure indicates significant differences between transporter transcription at various cytokine cocktail concentrations relative to the vehicle control, but the vehicle control expression is not shown on the graph. Would it be possible to add these bars (for the vehicle control)? This would make the significant differences easier to understand.

Figure 2: As for figure 1, control bars would be helpful.

Discussion:

The discussion is clear and relevant

Line 236: too many commas! This is true throughout the manuscript; please have a grammar edit.

Line 297: “Nonetheless” seems odd here. You could just start the sentence with “The concentrations of cytokines…”

Line 306: “effect” should be “effects”.

Line 370: Why do you say “(up to 1 ng/mL)”? This appears from the figure to not be the case for all of the transporters listed, although it is for some.

Supplemental Material

Figure S1: Please include the bars for the vehicle control.

References

Reference 2. Please fix the title format so it is not all in capital letters.

Comments on the Quality of English Language

The paper is clear and well-written. Please reduce the number of commas used in the manuscript and check the grammar overall; in a few places wording/grammar seem awkward.

Reviewer 3 Report

Comments and Suggestions for Authors

In their manuscript, the authors investigated the influence of important proinflammatory cytokines on the mRNA expression of important drug transporters expressed in freshly isolated primary human proximal tubular cells. The study addresses an important topic regarding the transcriptional regulation of important renal transport proteins, is carefully performed and the conclusions are supported by the presented results. I have just some minor points to remark.

1.       I suggest including some details regarding the investigated transport proteins into the introduction. This manuscript may be important not only for transporter experts but also for other scientist not experienced in drug transporter research. On the other hand, some transporter details included into the discussion section (e.g. lines 273 – 278) can be removed.

2.       Including a table summarizing the most significant effects of cytokines on mRNA expression would be helpful.

3.       It should be discussed how the expression data of the different donors fit to published transcriptome data (e.g. The Tabula Sapiens: A multiple organ, single cell transcriptomic atlas of humans – www.tabula-sapiens-potal.ds.czbiohub.org or the human expression atlas – www.ebi.ac.uk/gxa/home)

Minor:

Line 56: OATP = organic anion transporting polypeptides

Line 93: 4 °C

Lines 258 – 260: please include reference for this statement

Sometimes the authors mentioned “cocktails” and sometimes “cocktail” – please check

Round 2

Reviewer 1 Report

Comments and Suggestions for Authors

Accepted for publication.